# Integrating Analytical Hierarchical Process and Network Optimization Model to Support Decision-Making on Biomass Terminal Selection

**Shuva Gautam** [1,2,*] **, Luc LeBel** [1,2] **and Baburam Rijal** [1]

1 Department of Wood and Forest Sciences, Université Laval, Québec City, QC G1V 0A6, Canada
2 FORAC Research Consortium, Université Laval, Québec City, QC G1V 0A6, Canada
* Correspondence: shuva-hari.gautam@sbf.ulaval.ca

**Abstract:** Forest biomass is an appealing bioenergy feedstock due its renewability, availability and potential to stimulate local economies. It is, however, voluminous, with heterogenous fuel characteristics and uncertainties in its supply. The feasibility of a bioenergy facility is contingent on a secure supply of uniform feedstock; a terminal in the supply chain can be useful in this regard. Biomass can be treated in the terminal to meet quality specifications and stored to overcome seasonality and supply disruptions. Nonetheless, such terminals require a significant capital investment; thus, the decision to use a terminal needs to be made judiciously. The decision process must account for a diverse set of factors that influence the terminal's effectiveness. These include both quantitative and qualitative factors. The objective of this study is to develop a multi-criteria decision-making framework that takes quantitative and qualitative factors into consideration while selecting a terminal. The framework consists of analytical hierarchy process to analyze qualitative information, and a mixed-integer programming model to evaluate quantitative information including fuel quality (moisture content and thermal value). This hybrid framework was implemented in a case study. It proved to be an effective tool for identifying terminals with the highest potential to generate value for the bioenergy supply chain.

**Keywords:** log yard; forest biomass; terminal; moisture content; bioenergy; AHP; MIP





## 1. Introduction

Increased environmental awareness worldwide is giving way to the green economy [1]. Investment on renewable energy sources have significantly increased around the world, and new sources continue to be explored [2]. The forest industry is well-positioned to support this development [3]. There is an abundance of forest residues that could be directed towards energy generation; these include (i) tree tops and branches that are a byproduct of logging [4], (ii) bark, chips and sawdust that are a byproduct of lumber manufacturing process [5], (iii) underutilized species not desired by the conventional forest industry [6], (iv) trees damaged by fire, insects, wind or other types of disturbances [7], and (v) invasive woody materials [8]. These byproducts and residues need to be managed appropriately to maintain healthy forests, incurring significant costs for private and public agencies [9]. Utilizing residual biomass for energy production has benefits: it allows for forrest managers to generate value from material that would otherwise have incurred costs to manage, it stimulates local economies and it strengthens energy security.

Despite the abundance of forest biomass and the above-mentioned benefits, its financial feasibility for energy production can be challenging [10]. Geographically, it is distributed across a large area. It is a feedstock with low energy density and a high rate of incombustible material [11]. Inefficiency in supply chain logistics can rapidly make biomass procurement infeasible. Additionally, wood is highly heterogeneous in its properties, influencing energy yield [12]. Furthermore, these qualities change and even rapidly deteriorate

if not handled appropriately [13]. Quality, particularly moisture content, is important from a logistical perspective; a high level of moisture content makes the transportation of biomass inefficient. Moisture content is also an important parameter during the energy conversion phase, dictating the net energy yield from the feedstock. It is especially important for small- and medium-scale boilers that require it to be within a narrow range [14].

Incorporating a terminal in the bioenergy supply chain can be a potential solution to this challenge of meeting quality specifications [15,16]. Biomass can be treated in the terminal to improve its quality to a level specified by the customers. Furthermore, terminals allow for supply chains to overcome seasonality and uncertainties associated with supply and demand. However, a terminal adds significant cost to a supply chain with already-low profit margins [17,18]. The value generated by incorporating a terminal in the supply chain needs to outweigh the cost. The value generated by a terminal depends on how well the factors that influence its success are taken into consideration while choosing a location for the terminal [19,20]. Once the decision is made, it is not easily reversible, as the installation of a terminal requires a considerable amount of time and monetary investment [21].

There are numerous studies that propose quantitative methods to determine the optimal location for a forest biomass terminal. Gunnarsson [22] appled a large mixed-integer linear programming model and heuristics to support decisions on which terminals to use in a forest supply chain. Rauch [23] presented a mixed-integer linear programming model to determine the optimal terminal in terms of processing capacity and location to be utilized in the supply chain. A scenario analysis was carried out to determine the cost-optimal terminal. Väätäinen [24] presented a discrete-event simulation model to evaluate the performance of a supply chain with terminal at various distances from a CHP plant. Fernandez-Lacruz [25] developed a discrete event simulation model built in ExtendSim to evaluate the usefulness of terminals in supplying raw materials to CHP and biorefinery plants of varying sizes. Abasian [26] developed a two-stage stochastic optimization model to evaluate the profitability of including a terminal to an existing forest supply chain. Berg [27] proposed an integrated optimization model that simultaneously minimizes harvesting, transportation and terminal costs for round wood, logging residues and salvage harvest to identify cost-efficient locations for terminal establishment.

There are also qualitative approaches to identifying terminal locations. Van Dael [28] combined a multi-criteria decision analysis and GIS to identify potential locations for biomass valorization. A wide range of societal, environmental and technical criteria were taken into consideration. Macro-screening was followed by a micro-screening procedure to identify potential locations. Kühmaier [29] applied the analytical hierarchy process to identify terminal locations based on stakeholder preferences. First, suitable areas for terminal location were delineated, followed by the solicitation of expert opinions. The data were processed using AHP to develop a suitability index map from which potential sites were identified. The process allowed for a range of social and ecological criteria to be taken into consideration that were of concern to stakeholders.

As evident in the studies cited above, the methods proposed to determine the optimal location of a terminal can be classified as either qualitative or quantitative. Quantitative analyses generally take into consideration supply chain network parameters such as transportation distances, costs, market prices, volumes available, etc., in quantitative terms to determine the optimal location. However, this misses many elements that cannot readily be converted into numbers. Qualitative analyses take into consideration factors that are not easily quantifiable such as, availability of services, layout of the sites in consideration, supply security, level of support from the local government, etc. It is imperative that both qualitative and quantitative factors be taken into consideration in analyses to determine the optimal terminal location. Thus, the objectives of this study are to (i) propose a decision-making framework that takes into consideration both qualitative and quantitative factors in choosing an optimal site for a forest biomass terminal, and (ii) demonstrate the usefulness of the proposed framework through application in a case study.

## 2. Materials and Methods

The proposed decision support tool for selecting an optimal site for a forest biomass terminal takes into consideration both qualitative and quantitative elements. Given a set of terminals to choose from, analytical hierarchy process (AHP) was used to rank the options based on qualitative factors. Meanwhile, a mixed-integer programming (MIP) model ranked the terminals based on supply chain costs. A schematic of the overall decision-making framework for terminal selection is provided in Figure 1. As an output, the AHP procedure ascribed a score to each of the terminals under consideration. These scores were recognized as the benefits associated with the terminals. The greater the score, the higher the ranking of a particular terminal. The output of the optimization model was the determination of the cost associated with using each of the terminal. Subsequently, an analysis was carried out to identify the terminal with the highest benefit–cost ratio in the evaluation phase.

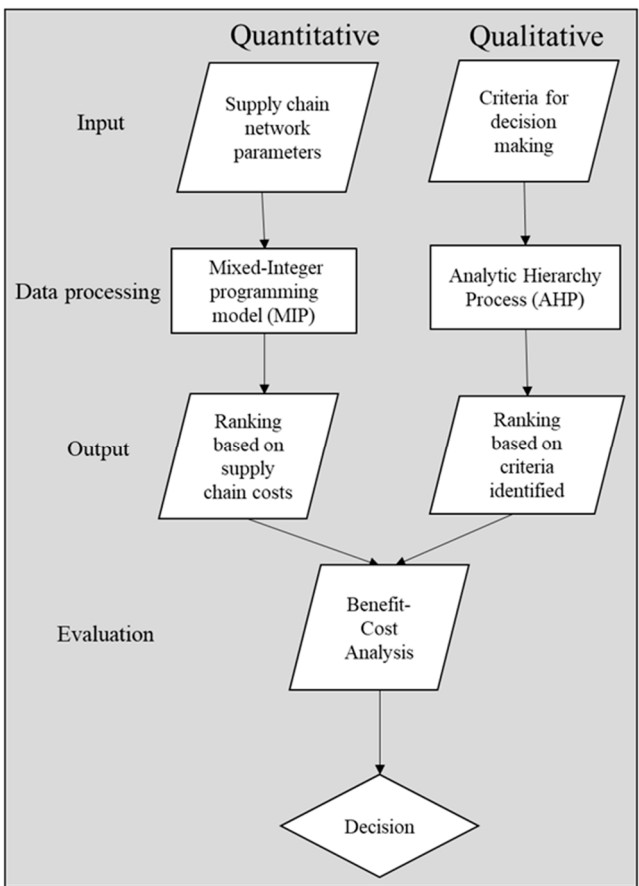

**Figure 1.** Proposed framework for terminal selection decision-making.

### 2.1. Analytical Hierarchy Process

The analytical hierarchy process (AHP) is a multi-criteria decision-making tool that allows for the simultaneous consideration of several but preferably qualitative criteria. AHP is based on "the principles of decomposition, comparative judgments, and synthesis of priorities" [30]. In the decomposition process, all possible alternative options (terminals in our case) that are to be made accessible for the supply chain are listed. The second step in this process is to assign a relative weight or score to each of the criteria associated with the specified alternative solution. The relative importance of each criterion is established through pairwise comparisons. The eigenvector of the scores for the matrix of alternatives and criteria are subsequently calculated. According to Saaty [30], the eigen vector approach

is the best approach to prioritize the alternatives. The following four steps provide an overview of the procedures in AHP.

Step I: Identification and listing of feasible alternatives and set criteria to be considered: these criteria are grouped in logical categories in such a way that they can be assigned relative importance values by pair. Ratings are attained through pairwise comparisons of the form "with respect to criterion A, alternative 1 is x times as desirable as alternative 2". Let there be m possible alternatives and n different criteria. For each criterion ($j^{th}$), a square matrix is made for all possible alternatives. We assign a score for each pair set in a matrix form by row for alternatives and by column for criteria. A relative score $\alpha_{ij}$ is assigned for $i^{th}$ alternative and $j^{th}$ criterion in such a way that $\sum_{j=1}^{n} \alpha_{ij} = 1$.

Step 2: For each criterion that are under consideration, a relative importance value is assigned for each pair. For example, relative importance score $a_{ij}$ is assigned by comparing the two alternatives $i^{th}$ and $k^{th}$ for the $j^{th}$ criterion. A square matrix of the pairwise comparison between two alternatives i and k for $j^{th}$ criterion is subsequently generated as shown in Equation (1). Each cell has the reciprocal property, i.e., $a_{ij} = \frac{1}{a_{ji}}$ as the score is based on pairwise comparisons. Such a square matrix is constructed for each of the criteria under consideration.

$$
\begin{pmatrix}
a_{11} & . & . & a_{1i} & a_{1j} & . & a_{1m} \\
a_{21} & a_{22} & . & . & . & . & a_{2m} \\
. & . & . & a_{ii} & . & . & . \\
a_{j1} & . & . & . & a_{jj} & . & a_{jm} \\
. & . & . & . & . & . & . \\
a_{m1} & a_{m2} & . & a_{mi} & a_{mj} & . & a_{mm}
\end{pmatrix}
\tag{1}
$$

Step 3. Numerical calculation is carried out to find eigenvector (ev). The squared matrix cells are normalized by row and eigenvector ($\alpha_i^*$) is calculated using the Equation (2).

$$
\alpha_i^* = \frac{\sum_{j=1}^{m} \alpha_{ij}}{\sum_{j=1}^{m} \sum_{i=1}^{m} \alpha_{ij}}
\tag{2}
$$

Subsequently, the eigenvectors are calculated. The matrix (Equation (1)) is squared for several iterations (n) until the differences between two consecutive eigenvectors are smaller than predefined *relative weight* $\leq \varepsilon$ (epsilon; a predefined value). Finally, a matrix showing the relative weights that has less than or equal to the specified number ($\varepsilon$) is retained (Equation (3)).

$$
\begin{pmatrix}
ev_1 \\
ev_2 \\
\ldots \\
ev_i \\
\ldots \\
ev_m
\end{pmatrix}
\tag{3}
$$

Next, a pairwise comparison of the alternatives (terminals) was made for each identified criterion. Thus, the above process of determining ev (until $\Delta$ ev $\leq \varepsilon$) was carried out for each of the criteria. The importance of each criterion to the overall decision depends on its share, which is the product of the criterion weight and the weight of the category to which the criterion belongs.

Step 4: Rankings: the ratings of the alternatives are combined with criteria shares into an overall rating for each investment alternative. The alternative with the highest overall rating is ranked the best choice, taking into account the relative importance of each criterion as well as the relative desirability of the alternatives with respect to each criterion.

### 2.2. Optimization Model

The optimization model developed is a mixed-integer programming (MIP) model that provides a cost-optimal decision on the quantity of forest biomass that should flow

through the network to meet customers' demands. The model sets, input data and decision variables are provided in Tables 1–3, respectively. The model takes into consideration the fluctuation in moisture content within biomass as it flows through the network. Material flow starts at cutblocks (subdivision of forests); each cutblock contains a known quantity of forest biomass which can be either (a) comminuted in the forest and directly transported to the customer facilities or, (b) transported to the terminal for storage without being comminuted. The model permits transportation of biomass to the customers only after quality requirements are met. In the terminal, there are two methods by which quality can be improved; it can be stored in the log yard for a several periods prior to being transported to customers, or it can be comminuted and stored inside a depot within the terminal, where quality can be better controlled. The latter option will incur a higher cost.

**Table 1.** Set notations used in the optimization model.

| Notation | Description |
| --- | --- |
| $H$ | Set of cutblocks from which biomass can be procured |
| $W$ | Set of terminals where biomass can be stored |
| $S$ | Set of depots where biomass can be stored |
| $C$ | Set of clients with demand for biomass |
| $P$ | Set of time periods |
| $A$ | Set of time periods in which biomass enter depots |

**Table 2.** Input parameters of the optimization model.

| Notation | Description |
| --- | --- |
| $b^i$ | Capital investment cost and terminal operation cost ($\cdot$year$^{-1}$) |
| $b^c$ | Comminution cost ($\cdot$ODt$^{-1}$) |
| $b^s$ | Stumpage fees paid to the government ($\cdot$gt$^{-1}$) |
| $b^l$ | Cost incurred to load biomass for transportation ($\cdot$gt$^{-1}$) |
| $b^u$ | Cost incurred to unload biomass after transportation ($\cdot$gt$^{-1}$) |
| $t^e$ | Total time taken to load and unload equipment for transportation (h) |
| $b^e$ | Payment rate to equipment transportation company ($\cdot$h$^{-1}$) |
| $r_{hp}$ | Period $p$ in which cutblock $h$ was harvested obtains a value of 0, 1 otherwise |
| $g_h$ | Length of road that requires upgrade when procuring from cutblock $h$ (km) |
| $b^r$ | Payment rate to upgrade roads ($\cdot$km$^{-1}$) |
| $b^h$ | Handling cost of material in the terminal ($\cdot$ODt$^{-1}$) |
| $t^v$ | Total time taken to load and unload a load of biomass from a truck (hr) |
| $b^t$ | Payment rate ($\cdot$h$^{-1}$) to trucking company |
| $o^t$ | Maximum payload (green tonne) |
| $d_{hc}$ | Distance (km) from cutblock $h$ to customer $c$ |
| $d_{hw}$ | Distance (km) from cutblock $h$ to terminal $w$ |
| $d_{wc}$ | Distance (km) from terminal $w$ to customer $c$ |

**Table 2.** *Cont.*

| Notation | Description |
|---|---|
| $d_{sc}$ | Distance (km) from depot $s$ to customer $c$ |
| $k_{hc}$ | Traveling speed (km·h$^{-1}$) from cutblock $h$ to customer $c$ |
| $k_{hw}$ | Traveling speed (km·h$^{-1}$) from cutblock $h$ to terminal $w$ |
| $k_{wc}$ | Traveling speed (km·h$^{-1}$) from terminal $w$ to customer $c$ |
| $k_{sc}$ | Traveling speed (km·h$^{-1}$) from depot $s$ to customer $c$ |
| $i^c$ | Inventory cost at terminal and depot |
| $v_h$ | Amount of biomass available(ODt) in cutblock $h$ |
| $d_{cp}$ | Demand of energy (GJ) by customer $c$ in period $p$ |
| $o_w$ | Storage capacity of biomass in terminal $w$ (gt) |
| $o_s$ | Storage capacity of biomass in depot $s$ (gt) |
| $m_{hp}$ | MC of biomass from cutblock $h$ in period $p$ in dry basis |
| $j^v$ | The higher heating value of biomass (GJ·t$^{-1}$) |
| $m_c^{max}$ | The maximum value of MC that can be transported to customer $c$ |
| $m_c^{min}$ | The minimum value of MC that can be transported to customer $c$ |
| $m_{ap}^{red}$ | Percentage reduction in MC in period $p$ of material that entered the depot in period $a$ |
| $n_c$ | Ratio between input of energy content of biomass and energy output |
| $q^w$ | Constant 2.447 to represent the latent heat of vaporization of water |
| $q$ | A small number |

**Table 3.** Decision variables of the mixed-integer programming model.

| Notation | Description |
|---|---|
| $F_{hcp}$ | Flow of biomass from cutblock $h$ to customer $c$ in period $p$ |
| $F_{hwp}$ | Flow of biomass from cutblock $h$ to terminal $w$ in period $p$ |
| $F_{hwcp}$ | Flow of biomass from terminal $w$ to customer $c$ in period $p$ of material from cutblock $h$ |
| $F_{hwsp}$ | Flow of biomass from terminal $w$ to depot $s$ in period $p$ of material from cutblock $h$ |
| $F_{hsa}$ | Flow of biomass from cutblock $h$ to depot $s$ in period $a$ |
| $F_{hscap}$ | Flow of biomass in period $p$ from depot $s$ to customer $c$ of material from cutblock $h$ that arrived in the depot in period $a$ |
| $I_{whp}$ | Inventory of biomass in terminal $w$ of material from cutblock $h$ in period $p$ |
| $I_{shap}$ | Inventory in period $p$ of biomass in depot $s$ of material from cutblock $h$ and arrived in period $a$ |
| $E_{hp}$ | 1 if biomass flows from cutblock $h$ in period $p$, 0 otherwise |
| $E_{hcp}$ | 1 if biomass flows from cutblock $h$ to customer $c$ in period $p$, 0 otherwise |
| $E_{hwcp}$ | 1 if biomass flows from terminal $w$ to customer $c$ in period $p$ of material from cutblock $h$, 0 otherwise |
| $E_{hscap}$ | 1 if biomass flows from depot $s$ to customer $c$ in period $p$ of material from cutblock $h$ and arrived in period $a$, 0 otherwise |

The objective function of the model is presented in Equation (4); capital investment and terminal operation cost are taken into consideration by the first component. The

second element captures the cost incurred when biomass is comminuted in the cutblock and transported to biorefineries. The stumpage value and the cost incurred when loading biomass for direct transportation to customer facilities is captured by the third element. The cost of transporting, comminution and loading equipment when biomass is transported from the cutblocks directly to customer facilities is captured by the fourth and fifth element. Road maintenance costs incurred to access cutblocks are captured by the sixth element. Stumpage, loading and unloading costs incurred when biomass is transported from a cutblock to the terminal are accounted for by the seventh element. Comminution and loading costs incurred when biomass is transported from the terminal to customer facilities are taken into consideration by the eighth and ninth elements. Comminution and handling costs incurred when biomass is processed in the log yard and sent to a depot for storage within the terminal are taken into consideration by the tenth and eleventh elements. The cost of loading biomass for transportation to the customer facilities is taken into consideration by the twelfth element. Costs of transporting biomass are taken into consideration by thirteenth through to sixteenth elements. Inventory costs incurred in the terminal and in the depot are accounted for by the fourteenth and fifteenth elements.

Minimize cost =

$$b^i + \sum_{h \in H} \sum_{c \in C} \sum_{p \in P} b^c * F_{hcp} + \sum_{h \in H} \sum_{c \in C} \sum_{p \in P} \left( b^s + b^l \right) \left( F_{hcp} + \left( \frac{m_{hp}}{100} * F_{hcp} \right) \right) \tag{4}$$

$$+ \sum_{h \in H} \sum_{w \in W} \sum_{c \in C} \sum_{p \in P} \left( \frac{d_{hw}}{k_{hw}} + t^e \right) b^e * E_{hcp} + \sum_{h \in H} \sum_{w \in W} \sum_{p \in P} r_{hp} \left( \frac{d_{hw}}{k_{hw}} + t^e \right) b^e * E_{hp}$$

$$+ \sum_{h \in H} \sum_{w \in W} \sum_{c \in C} \sum_{p \in P} r_{hp} * g_h * b^r * E_{hp} + \sum_{h \in H} \sum_{w \in W} \sum_{p \in P} \left( b^s + b^l + b^u \right) \left( F_{hwp} + \left( \frac{m_{hp}}{100} * F_{hwp} \right) \right)$$

$$+ \sum_{h \in H} \sum_{w \in W} \sum_{c \in C} \sum_{p \in P} b^c * F_{hwcp} + \sum_{h \in H} \sum_{w \in W} \sum_{c \in C} \sum_{p \in P} b^l \left( F_{hwcp} + \left( \frac{m_{hp}}{100} * F_{hwcp} \right) \right)$$

$$+ \sum_{h \in H} \sum_{w \in W} \sum_{s \in S} \sum_{p \in P} b^c * F_{hwsp} + \sum_{h \in H} \sum_{w \in W} \sum_{s \in S} \sum_{p \in P} b^h \left( F_{hwsp} + \left( \frac{m_{hp}}{100} * F_{hwsp} \right) \right)$$

$$+ \sum_{h \in H} \sum_{s \in S} \sum_{c \in C} \sum_{a \in A} \sum_{p \in P} b^l \left( F_{hscap} + \left( \frac{m_{hp}}{100} * F_{hscap} \right) \right)$$

$$+ \sum_{h \in H} \sum_{c \in C} \sum_{p \in P} \left( \left( \frac{d_{hc}}{k_{hc}} + t^v \right) * b^t \right) * \left( \left( \left( F_{hcp} * \frac{m_{hp}}{100} \right) + F_{hcp} \right) / o^t \right)$$

$$+ \sum_{h \in H} \sum_{w \in W} \sum_{p \in P} \left( \left( \frac{d_{hw}}{k_{hw}} + t^v \right) * b^t \right) * \left( \left( \left( F_{hwp} * \frac{m_{hp}}{100} \right) + F_{hwp} \right) / o^t \right)$$

$$+ \sum_{h \in H} \sum_{w \in W} \sum_{c \in C} \sum_{p \in P} \left( \left( \frac{d_{wc}}{k_{wc}} + t^v \right) * b^t \right) * \left( \left( \left( F_{hwcp} * \frac{m_{hp}}{100} \right) + F_{hwcp} \right) / o^t \right)$$

$$+ \sum_{h \in H} \sum_{w \in W} \sum_{s \in S} \sum_{c \in C} \sum_{a \in A} \sum_{p \in P} \left( \left( \frac{d_{sc}}{k_{sc}} + t^v \right) * b^t \right) * \left( \left( \left( F_{hscap} * \frac{m_{hp} * m_{ap}^{red}}{100} \right) + F_{hscap} \right) / o^t \right)$$

$$+ \sum_{w \in W} \sum_{h \in H} \sum_{p \in P} i^c * I_{whp} + \sum_{s \in S} \sum_{h \in H} \sum_{a \in A} \sum_{p \in P} i^c * I_{shap}$$

The constraints of the model are presented in Equations (5)–(31). The binary variables and continuous variables are linked by Equations (5) and (6). The binary variable becomes 1 even if a small amount of biomass is procured from the cutblock. Equations (7) and (8) ensure that all available biomass is procured in subsequent periods. Equation (9) ensures that the volume procured in a cutblock does not surpass the total available. The model fulfills demand in each of the periods (Equation (10)).

Binary variables are used to ensure that moisture content of biomass is within the range specified by customers. When even a small amount of biomass flows to a customer facility from different sources, a binary value of 1 is retained. Binary variables are linked with continuous variables using Equations (11)–(16) and Equations (17)–(22) provide the moisture range. Equations (23)–(26) are flow conservation constraints for the terminal. Biomass arrival and departure periods in the terminal were tracked using two sets of time periods. Consistency between the two time periods is maintained using Equation (27). Terminal capacities are specified using Equations (28) and (29). Equation (30) defines the binary variables and Equation (31) is non-negativity constraint. The MIP model was coded in AMPL modelling language and solved using CPLEX 12.5 in a 2.8 GHz PC with 32 GB RAM. The optimality gap was set to within 1 percent and the time limit for computation was fixed at 10,000 s.

$$E_{hp} \leq \sum_{c \in C} F_{hcp} + \sum_{w \in W} F_{hwp} \qquad \forall\, h, p \tag{5}$$

$$E_{hp} \geq \left( \sum_{c \in C} F_{hcp} + \sum_{w \in W} F_{hwp} \right) * Q \qquad \forall\, h, p \tag{6}$$

$$\sum_{p \in P} v_h * E_{hp} = \sum_{c \in C} \sum_{p \in P} F_{hcp} + \sum_{w \in W} \sum_{p \in P} F_{hwp} \qquad \forall\, h \tag{7}$$

$$\sum_{p \in P} E_{hp} \leq 1 \qquad \forall\, h \tag{8}$$

$$\sum_{c \in C} \sum_{p \in P} F_{hcp} + \sum_{w \in W} \sum_{p \in P} F_{hwp} \leq V_h \qquad \forall\, h \tag{9}$$

$$\sum_{h \in H} \left( j^v - m_{hp} j^v - q^w m_{hp} \right) F_{hcp} + \sum_{h \in H} \sum_{w \in W} \left( j^v - m_{hp} j^v - q^w m_{hp} \right) F_{hwcp}$$
$$+ \sum_{h \in H} \sum_{s \in S} \sum_{a \in A} \left( J^v - m_{hp} j^v - q^w m_{hp} \right) F_{hscap} \qquad \geq \frac{d_{cp}}{n_c} \qquad \forall\, c, p \tag{10}$$

$$q * F_{hcp} \leq E_{hcp} \qquad \forall\, h, c, p \tag{11}$$

$$F_{hcp} \geq E_{hcp} \qquad \forall\, h, c, p \tag{12}$$

$$q * F_{hwcp} \leq E_{hwcp} \qquad \forall\, h, w, c, p \tag{13}$$

$$F_{hwcp} \geq E_{hwcp} \qquad \forall\, h, w, c, p \tag{14}$$

$$q * F_{hscap} \leq E_{hscap} \qquad \forall\, h, s, c, a, p \tag{15}$$

$$F_{hscap} \geq E_{hscap} \qquad \forall\, h, s, c, a, p \tag{16}$$

$$m_{hp} E_{hcp} \leq m_c^{max} \qquad \forall\, h, c, p \tag{17}$$

$$m_{hp} E_{hcp} \geq m_c^{min} \qquad \forall\, h, c, p \tag{18}$$

$$m_{hp} E_{hwcp} \leq m_c^{max} \qquad \forall\, h, w, c, p \tag{19}$$

$$m_{hp} E_{hwcp} \geq m_c^{min} \qquad \forall\, h, w, c, p \tag{20}$$

$$m_{hp} E_{hscap} \leq m_c^{max} \qquad \forall\, h, s, c, a, p \tag{21}$$

$$m_{hp} E_{hscap} \geq m_c^{min} \qquad \forall\, h, s, c, a, p \tag{22}$$

$$I_{whp} = I_{w,h,p-1} + F_{hwp} - \sum_{c \in C} F_{hwcp} - \sum_{s \in S} F_{hwsp} \; \forall\, w, h \tag{23}$$

$$\sum_{c \in C} F_{hwcp} - \sum_{s \in S} F_{hwsp} \leq I_{whp} \; \forall\, w, p, h \tag{24}$$

$$I_{shap} = I_{s,h,a,p-1} + F_{hsa} - \sum_{c \in C} F_{hscap} \; \forall\, s, h, a, p \tag{25}$$

$$\sum_{c\in C} F_{hscap} \le I_{shap} \quad \forall \; p,h,s,a \tag{26}$$

$$\sum_{w\in W} F_{hwsp} = F_{hsa} \; \forall \; h,s, \; p=a \tag{27}$$

$$\sum_{h\in H} I_{whp} \le o_w \quad \forall \; w,p \tag{28}$$

$$\sum_{a\in A}\sum_{h\in H} I_{shap} \le o_s \quad \forall \; s,p \tag{29}$$

$$E_{hp}, \; E_{hcp}, \; E_{hwcp}, \; E_{hscap} \in \{0,1\} \tag{30}$$

$$F_{hcp}, \; F_{hwp}, \; F_{hwcp}, \; F_{hwsp}, \; F_{hsa}, \; F_{hscap}, \; I_{whp}, \; I_{shap} \; \ge 0 \quad \forall \quad h,w,s,c,p,a \tag{31}$$

### 2.3. Case Study Description

The proposed hybrid decision-making framework was applied to the case of a forestry organization in Quebec, Canada that manages approximately 118,000 ha of forest land, which are scattered in five discontiguous regions. They harvest and supply approximately 150,000 m$^3$ of wood annually to surrounding mills. All these forests in this area contain mixed hardwood species. When a forest is harvested, the higher-quality materials are sold to veneer mills, sawmill and pulpmills. The remaining biomass with a diameter below 9 cm is left on the ground as logging residues. As a new business venture, the organization had signed agreements with several small-scale commercial customers to supply this biomass to be used as feedstock for bioenergy production. The customers included hospitals, schools and commercial buildings with boilers that can generate heat and electricity using biomass. Four potential alternative sites were considered for locating a biomass terminal (Figure 2).

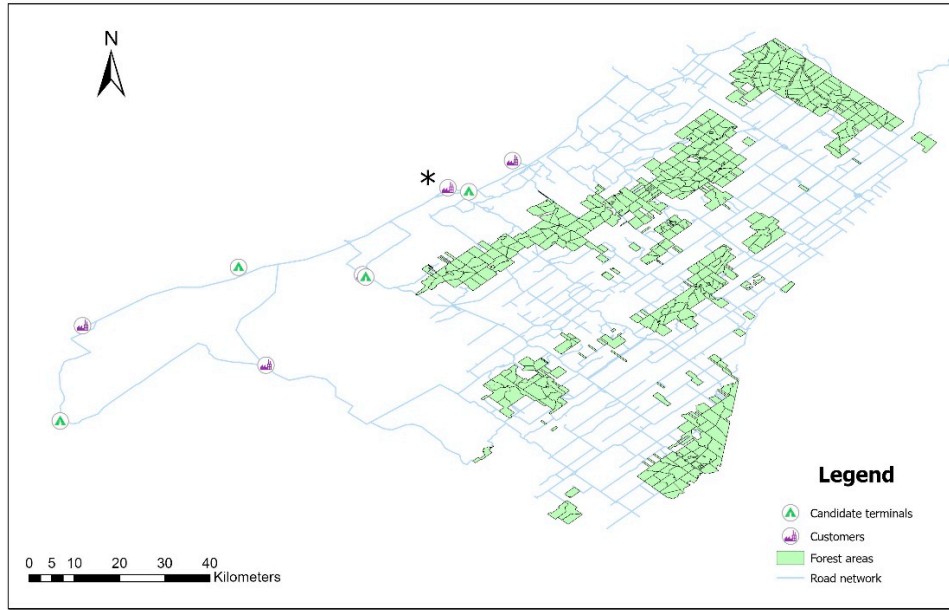

**Figure 2.** Case study area map showing the forest supply regions, candidate terminal locations, location of customers' facilities and the road network. * represents the terminal selected using the proposed framework.

Inventory data for the case study were obtained from the forestry organization. Given that biomass procurement is only feasible in cutblocks that are harvested for conventional products, the potential supply area was limited to wood procurement plans of the forestry

organization. The quantity of biomass available for the bioenergy supply chain was subsequently determined using the following Equations (32)–(33) [31].

$$y_{wood} = \beta_{wood1} D^{\beta wood2} H^{\beta wood3} + e_{wood} \tag{32}$$

$$y_{bark} = \beta_{bark1} D^{\beta bark2} H^{\beta bark3} + e_{bark} \tag{33}$$

$$y_{stem} = \hat{y}_{wood} + \hat{y}_{bark} + e_{stem} \tag{34}$$

where $y_i$ is the quantity of dry biomass in kilograms; D is the diameter at breast height (dbh) in centimetres; H is the height in metres. $\beta_i$ and $e_i$ are constants provided in [31] for different tree species. The distances between each cutblock, potential terminal sites and customers were determined using GIS software. The quantity demanded per period by each customer and other quantitative parameters required for the MIP model were obtained from the forestry organization. Four potential sites were identified by the company for operating a terminal. The decision-making framework presented in Figure 1 was implemented to make a choice among the four sites.

Two methods were used to collect qualitative data for analysis using AHP: (i) a literature search and (ii) consultation with practitioners. For the first method, a literature search was conducted, focusing strictly on studies carried out on forest biomass terminals. However, this is by no means an attempt to develop an exhaustive list of criteria for terminal selection. The objective was to demonstrate the utility of our proposed method. Documents with significant contribution in the criteria selection procedure included: [15,19,21,32–34]. criteria used for the AHP procedure are detailed in Box 1.

**Box 1.** Criteria that were input for the AHP procedure.

1. Terminal setup relates to the total area, shape, location and aspect of the site in consideration. Total area dictates the congestion level and the ease of carrying out daily operations in the terminal. It can also be a limiting factor in future expansion plans. The shape of the terminal will also impact the ease of operation, potentially impacting handling costs. Location and aspect have impact the wind pattern and exposure to sunlight. These factors will be important in improving biomass quality.
2. Proximity to forest products manufacturers: Although the primary feedstock source is the surrounding forests, having forest products manufacturers in the vicinity can offer a cheaper option regarding supply. In some instances, the by-products may have already been dried, providing an additional advantage. Additionally, a terminal in close proximity to other forest products manufacturers could mean that biomass procurement costs could be lowered through resource-sharing.
3. Infrastructure in place: At a minimum, a terminal will require a balance to measure biomass, a shed to protect biomass from precipitation, a paved area to place the biomass so that dirt does not get mixed in with the feedstock. Investments will need to be made to install these infrastructures if they are not already in place.
4. Access to services: Access to electricity, gas, water, and sewage will be needed to ensure an effective and safe working environment. This criterion includes other factors, such as distance from hospital, fire and police station.
5. Labour availability: Successful operation of the terminal will depend on the availability of a skilled work force. In certain rural areas, the availability of labour may be scarce, while in other areas t hismay not be an issue.
6. Proximity to railroad: Truck is usually the primary mode of transportation. However, if growth is planned in the future, access to a rail network will be essential to improve efficiency in transportation.

Additional criteria were identified, e.g., closeness to customers, closeness to wood supply, government subsidies. However, these criteria will have a direct effect on the cost. Any criteria that can be measured in terms of (direct) cost were taken into consideration by the MIP model. Thus, these criteria were not included in the AHP procedure.

## 3. Results

The first step of the AHP process was to generate the relative ranking of the different criteria. This was generated through a pairwise comparison. This process involved the wood procurement staff members from the company interested in installing a terminal. The results of these scorings are shown in Table 4; the consistency ratio was calculated to be 0.048 < recommended threshold of 10%. For this particular case, it was found that the proximity of forest products manufacturers was the most important criterion, followed closely by terminal setup. On the other hand, proximity to railroad was judged to be the least important factor.

**Table 4.** The relative ranking of terminal selection criteria.

| Criteria | Eigenvector |
|---|---|
| Proximity of forest products manufacturers | 0.2845 |
| Terminal setup | 0.2736 |
| Labour availability | 0.1463 |
| Access to services | 0.1415 |
| Infrastructure in place | 0.0932 |
| Proximity to railroad | 0.0607 |

Next, a pairwise comparison of the terminals was made for each criterion. This calculation generated a ranking of the terminals under the 6 criteria (Box 1). This matrix of eigenvectors was subsequently multiplied by Table 4, generating a relative ranking of the different terminals (Table 5, second column). Site 1 was found to be the most interesting location for operating a terminal, followed closely by Site 3.

**Table 5.** The relative ranking and the annual operating cost of each terminal in consideration obtained using the mixed-integer programming model.

| Site | Eigenvector | Cost ($) |
|---|---|---|
| 1 | 0.3208 | 317,490 |
| 2 | 0.1281 | 367,343 |
| 3 | 0.3130 | 297,493 |
| 4 | 0.2380 | 316,304 |

The next stage included incorporating the results of the optimization model in the decision-making process. The results of the MIP model are presented in Table 5, column 3. For each terminal, the cost represents the yearly expenditure incurred in fulfilling demand. Based solely on the cost, Site 3 would be the preferred location for operating a terminal, followed by Site 4.

In the final step of the procedure, the cost was normalized, and benefit–cost ratio was calculated for each terminal. The normalization of the cost was done by summing the costs and determining the contribution (ratio) of each terminal. The benefit–cost ratio was subsequently obtained by dividing the eigenvectors of the terminals (Table 5, column 2) by the normalized costs. Site 3 displayed the highest benefit–cost ratio of 1.37 (indicated as * in figure Figure 2), followed by Site 1 (1.31), Site 4 (0.98) and Site 2 (0.45).

## 4. Discussion

The application of the proposed framework to the case study demonstrated the usefulness of integrating both qualitative and quantitative data to support decision-making on terminal location. Although the advice for the decision-maker was to choose site 3, others may have been recommended if only one of the two methods was utilized. The benefit cost analysis shows that sites 1 and 3 were superior to sites 2 and 4. The distinction between sites 1 and 3 was not as pronounced. Relying solely on AHP analysis would have led to the recommendation of site 1 due to its higher ranking. On the other hand, relying

exclusively on MIP would have led to the recommendation of Site 3, as this incurred the lowest cost. However, a slight change in the MIP model parameters could have easily led to the recommendation of either site 1 or 4. The integration of both models provided a robust analytical tool to narrow down the list to only the most interesting options, sites 1 and 3 in our case.

The primary motivation to develop the proposed model was to help the forestry organization described in the case study to identify the most suitable terminal in terms of economic, social and geographic attributes. The organization's management expressed great satisfaction with the results obtained using the proposed modeling framework. Decision on terminal location are not trivial and prudent organizations will evaluate all the factors listed in the modelling framework. However, this is rarely performed in an integrated manner, and generally carried out sequentially [29,35]. Making these decisions sequentially can lead to a suboptimal decision. For example, if the MIP model was used first to narrow down the options, only sites 3 and 4 would have been considered for further evaluation. Site 1 would have been eliminated for further evaluation, although we know in this case that it was quite favorable from a qualitative perspective.

Personnel at the forestry organization particularly appreciated its ability to simultaneously take into consideration numerous factors. The model incorporated their managerial inputs, location factors as well as biomass quality into the site selection process. Although feedstock quality has been demonstrated to be a critical factor that influences profitability of biomass supply chain [18,36], it is generally not considered in such analyses. As such, one of the unique aspects of this proposed framework is that biomass quality (moisture content and heat value) can be explicitly taken into consideration in the terminal location decision-making.

Most of the data used in the case study were provided by the forestry organization who expressed a high level of confidence in the data. A deterministic approach was, therefore, sufficient to provide answers to specific questions of the case study. Future work that is geared towards making general recommendations should consider using stochasticity of input parameters. Several stochastic models have been published, which can be adopted to the proposed modeling framework. Abasian [26] considered uncertainty in the demand and price of final product in their optimization model. Several models have been proposed to consider the uncertainty associated with supply [5,37]. For the AHP analysis, fuzzy set theory is recommended for certain criteria to determine the suitability of a terminal [29]. Auer [38] provide a more comprehensive list of sources of uncertainties and methods that can be incorporated in our decision-making framework.

The focus of this study was on biomass supply terminal and its logistics. In other cases, it may be essential to consider a range of environmental, social and economic factors to address the concerns of different stakeholders, such as local governments, environmental agencies and industries. Studies have highlighted the range of factors that may need to be evaluated prior to installing a terminal [19–21]. Such information could easily be incorporated in our modeling framework, either in the qualitative or quantitative form. Furthermore, the framework can also be readily applied to other types of terminals that manage a range of feedstocks for the forest products industry.

## 5. Conclusions

This study proposes a decision-making framework that considers both qualitative and quantitative factors when making recommendations regarding the optimal site for a forest biomass terminal location. The case study demonstrated that the proposed framework proved to be a convenient and effective tool for practitioners to support their decision-making process. It was shown that using only one of the methods (either AHP or MIP) could easily lead to a suboptimal decision. Once made, the decision cannot be reversed without severe financial repercussions. These terminals require investments ranging from several hundred thousand dollars to millions. A suboptimal decision at this point could lead to a poor performance and inconvenience for many years into the future.

The focus in the case study was on biomass terminal and its logistics; as a result, the criteria were developed, and the results represent the forest organization's viewpoint. Developing environmental, social and economic criteria to address concerns of a range of stakeholder may lead to a different ranking of the potential terminals. Future work should focus on using the proposed framework in such scenarios and incorporating the uncertainty attributes of data.

**Author Contributions:** Conceptualization, S.G. and L.L.; methodology, S.G.; software, S.G. and B.R.; validation, S.G., L.L. and B.R.; formal analysis, S.G.; investigation, S.G.; resources, S.G.; data curation, S.G.; writing—original draft preparation, S.G.; writing—review and editing, B.R. and L.L.; visualization, S.G.; project administration, L.L.; funding acquisition, L.L. All authors have read and agreed to the published version of the manuscript.

**Funding:** NSERC/Industrial Research Chair for Smart Supply Systems within the connected forest value chain and discovery grant program (RGPIN/05602-2018).

**Institutional Review Board Statement:** Not applicable.

**Informed Consent Statement:** Not applicable.

**Data Availability Statement:** Not applicable.

**Acknowledgments:** CIRRELT: FORAC, Quebec Federation of Forestry Co-operatives.

**Conflicts of Interest:** The authors declare no conflict of interest.

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
