# Peer review of "Integrating Analytical Hierarchical Process and Network Optimization Model to Support Decision-Making on Biomass Terminal Selection"

_forests, doi:10.3390/f13111898_

Round 1

Reviewer 1 Report

 This study proposes a decision-making framework that considers qualitative and 100 quantitative factors in choosing an optimal site for a forest biomass terminal. 

1. AHP is a mature and straightforward method, it is enough to give a framework of a process, with no need to introduce each process in detail.

2. You may need a map to show your result of the case study, which shows the final site.

3. Please explain the solution methods of MIP.

Author Response

  1. AHP is a mature and straightforward method,it is enough to give a framework of a process, with no need to introduce each process in detail.

We would like to thank the reviewer for investing valuable time reviewing the manuscript. The method explaining AHP has been reviewed, the text is more concise now and Table 1 was removed.

  1. You may need a map to show your result of the case study,which shows the final site.

An asterisk has been placed on the map to indicate the final site chosen. This is also referred in the results section, see lines 309-310.

  1. Please explain the solution methods of MIP.

We have now provided additional information regarding the software used to code the MIP, hardware used for the experiment and also criteria for obtaining the solutions. Please see lines 232-235.

Reviewer 2 Report

The abstract contains all necessary items.

The introduction it is clear and well written. There is a good set of bibliographic citations that corroborate the defense of the theme.

The Materials and Methods are quite detailed and reproducible by other researchers. A series of equations is presented. The objective function of the model presented in Equation 4 it's not easy to interpret. I suggest put numbering in each component presented.

In results

At lines 299 and 300, where it is written: Based solely on the cost, Site 3 would be the preferred location for operating a terminal, followed by Site 1. 

I suggest correcting by "Based solely on the cost, Site 3 would be the preferred location for operating a terminal, followed by Site 4". See that 316.304 (site 4) is less than 317.490 (site 1).

Author Response

The abstract contains all necessary items.

The introduction it is clear and well written. There is a good set of bibliographic citations that corroborate the defense of the theme.

The Materials and Methods are quite detailed and reproducible by other researchers. A series of equations is presented. The objective function of the model presented in Equation 4 it's not easy to interpret. I suggest put numbering in each component presented.

We appreciate the valuable time invested by the reviewer to provide us feedback. The objective function of the model presented in Equation 4 is now divided into smaller components with each receiving numbering to facilitate readers. They are accordingly referenced in text, please see lines 195- 216

In results

At lines 299 and 300, where it is written: Based solely on the cost, Site 3 would be the preferred location for operating a terminal, followed by Site 1. 

I suggest correcting by "Based solely on the cost, Site 3 would be the preferred location for operating a terminal, followed by Site 4". See that 316.304 (site 4) is less than 317.490 (site 1).

Thank you for catching this error, this has been corrected in the text, see lines 303-304